# Explaining Tabular Foundation Model Differences Through Meta-Features

**Markus Herre** [1]   **Andrej Tschalzev** [2]   **Sascha Marton** [1]   **Christian Bartelt** [1]

## Abstract

With the rise of tabular foundation models alongside traditional models still performing well on many tasks, choosing the right model for a tabular dataset remains difficult. We investigate whether dataset meta-features can explain performance gaps between model families on tabular prediction tasks. Using the TabArena benchmark results, we analyze dataset-level performance gaps and relate them to model-agnostic meta-features. After strict statistical tests with false discovery control, we find that (1) for neural network vs. tree gaps, no meta-feature survives false discovery control, (2) for non-foundation vs. foundation model gaps, one association is robust but does not generalize when tested in leave-one-dataset-out prediction, and (3) for TabICLv2 vs. TabPFN-2.6, one robust association also improves held-out prediction. Furthermore, we conduct a leave-one-dataset-out analysis and find that meta-feature predictors fail to improve meaningfully over a simple baseline. Overall, our results show the heterogeneity of tabular datasets and that global meta-feature approaches are not robust enough to offer explanations on the 51 TabArena datasets.

## 1. Introduction

Choosing the best model for a tabular data task has become increasingly challenging as foundation models begin to challenge long-standing assumptions about which model families work best and why. Recent work shows that, while gradient-boosted decision trees (GBDTs) remain competitive, deep tabular models and foundation models dominate benchmarking studies (Erickson et al., 2025; Hollmann et al., 2025; Ye et al., 2025). From a practitioner's point of view, the variety of model families implies a routing problem: given a new dataset, which model family is likely to

perform best and are there aspects of related datasets (meta-features) that can be used to generalize from benchmark performance to a new dataset?

Dataset meta-features are a natural candidate for this routing problem. Meta-learning has previously used task meta-features to transfer knowledge across datasets (Vanschoren, 2019). Previous benchmark studies on tabular data found that some dataset meta-features help characterize when neural networks (NNs) or boosted trees are favored (McElfresh et al., 2023). At the same time, recent work shows that these conclusions were strongly shaped by the evaluation setup and vanish under improved protocols (Tschalzev et al., 2025). While previous benchmarks were biased, the recently released TabArena benchmark addresses these concerns through its manual curation and controlled evaluation setup (Erickson et al., 2025).

This motivates revisiting the plausibility of meta-feature routing using a benchmark with strong baselines and an evaluation protocol designed to reduce bias. We therefore ask whether global dataset meta-features can explain or predict model-family performance gaps in TabArena. We compare three types of models and families: (1) non-foundation tabular models (Non-TFMs) vs. tabular foundation models (TFMs), (2) TabICLv2 (Qu et al., 2026) vs. TabPFN-2.6 (Grinsztajn et al., 2026), and (3) NNs vs. tree-based models. In particular, we investigate two distinct questions: First, can meta-features explain performance differences between model families under statistical association analysis? Secondly, can meta-features be used for routing, i.e. predict which model family to use on an unseen dataset? Our results demonstrate that a meta-feature may exhibit significant explanatory power for the first question while still failing to generalize effectively for the second.

Across the 51 datasets and hundreds of candidate meta-features, only two meta-features survive our robust association screen. For the Non-TFM vs. TFM comparison, one meta-feature that captures the asymmetry in the per-feature entropy distribution yields a robust but only descriptive association. For TabICLv2 vs. TabPFN-2.6, the retained meta-feature captures the typical mutual predictability between features and is the only meta-feature that also improves held-out gap prediction. These findings suggest that global meta-features provide limited routing evidence overall.

[1]Clausthal University of Technology, Clausthal-Zellerfeld, Germany [2]University of Mannheim, Mannheim, Germany. Correspondence to: Markus Herre <markus.herre@tu-clausthal.de>.

*Proceedings of the $2^{nd}$ ICML Workshop on Foundation Models for Structured Data*, Seoul, South Korea. 2026. Copyright 2026 by the author(s).

## 2. Related Work

**Meta-features in tabular benchmarks.** The closest prior evidence for meta-feature routing comes from McElfresh et al. (2023), who compared 19 algorithms across 176 OpenML classification datasets and analyzed PyMFE meta-features (Alcobaça et al., 2020). Their study shows that NN vs. GBDT performance differences are associated with dataset properties, in particular size and distributional irregularity. The benchmarks TALENT (Ye et al., 2025) and OmniTabBench (Jiang et al., 2026) broaden this picture by studying larger collections of tabular datasets and reporting meta-feature and heterogeneity analyses across modern deep tabular methods, GBDTs, NNs, and foundation models. However, these settings rely on broader public-repository collections rather than a manually curated benchmark. Furthermore, others mostly emphasize descriptive association rather than generalization to unseen datasets (Uddin & Lu, 2024). Moreover, even when generalization was studied (McElfresh et al., 2023; Shmuel et al., 2024; Zhang et al., 2024), simple no-meta-feature baselines were often missing. A notable exception is the data-scarce regime of the PMLB-mini suite (Knauer et al., 2024). They benchmark AutoML and tabular foundation models against an L2-regularized logistic-regression baseline and use PyMFE meta-features to study when complex methods help, reporting that simple meta-features are weak predictors while complexity- and distribution-related descriptors carry more signal. Finally, even though previous approaches often used many datasets, this broader coverage required sacrifices in the evaluation design and reduced the auditability and external validity of downstream claims (Tschalzev et al., 2025).

**How our work differs.** We use the TabArena benchmark (Erickson et al., 2025), which, compared to prior work, offers a greatly improved and controlled evaluation setup: manually curated IID tasks, standardized implementations, nested, repeated cross-validation, extensive hyperparameter optimization (HPO), and foundation methods. All results were released allowing follow-up work on meta-feature analysis. This lets us ask whether global meta-features support reliable conclusions under a controlled modern protocol. Furthermore, we conduct a holdout evaluation to test the predictiveness of meta-features, and unlike most previous studies we (1) include simple mean/mode predictor baselines, and (2) include control feature models.

## 3. Methodology

**Study design.** We aim to explain performance differences between tabular model families (see Appendix A.3) using dataset-level properties. This requires three ingredients: a target that measures performance differences, a set of candidate explanatory dataset meta-features, and a standardized procedure for screening, validating, and evaluating these meta-features under statistical uncertainty. We conduct a predefined set of comparisons using the TabArena benchmark (Erickson et al., 2025). A more detailed description of TabArena is reported in Appendix A.1.

**Performance gap construction.** TabArena evaluates each dataset over multiple predefined train/validation/test splits. We use the term family for any predefined set of one or more methods. For a given split $s$, we select the best-performing method within each family $\mathcal{F}_A$ and $\mathcal{F}_B$ by validation error and evaluate the selected models on the held-out test set. Let $\tilde{e}_A(D, s)$ and $\tilde{e}_B(D, s)$ denote their normalized test errors (Equation (2), Appendix A.2). The dataset-level performance gap is the mean over all $|S|$ splits,

$$\Delta(D) = \frac{1}{|S|} \sum_{s \in S} \left[ \tilde{e}_A(D, s) - \tilde{e}_B(D, s) \right] \qquad (1)$$

Positive $\Delta(D)$ means that family $\mathcal{F}_B$ achieves lower error on dataset $D$. Averaging over splits is essential because folds from the same dataset share training data, so treating them as independent would inflate the effective sample size. Moreover, averaging over splits reduces variance, which is necessary to obtain reliable measurements.

**Meta-feature extraction and preprocessing.** All meta-features are computed from training data only to prevent any test set leakage. We extract three groups of meta-features: (a) *basic* properties, such as dataset size, feature composition, and missingness, (b) *redundancy* meta-features based on pairwise feature correlations, and (c) a large set of meta-features using the *PyMFE* library (Alcobaça et al., 2020), covering statistical, information-theoretic, and model-based summaries. We apply four preprocessing steps: (1) replace infinite values with NaN, (2) remove features with more than 20% missing values, (3) remove near constant features, and (4) retain at most one member from each group of mutually correlated features whose absolute Spearman correlation exceeds 0.95, retaining the member with fewer missing values and, as a tiebreaker, more unique values.

**Association screening.** We ask whether individual meta-features are systematically associated with dataset-level performance gaps across datasets. For each feature we compute the Spearman rank correlation with the dataset level gap $\Delta(D)$, using only the datasets for which both the feature and $\Delta(D)$ are observed. A two-tailed $p$-value is derived from the asymptotic $t$-distribution approximation for each correlation. To control for the large number of simultaneous tests (multi-hypothesis testing), we apply the Benjamini-Hochberg (BH) procedure at false discovery rate (FDR) level 0.05 (Benjamini & Hochberg, 1995). A feature passes our reporting rule only if it subsequently achieves bootstrap sign consistency of at least 0.90, defined as the fraction of 500 dataset level bootstrap resamples in which the Spearman correlation shares the sign of the point estimate (Efron,

*Figure 1.* Association pipeline for the general meta-feature matrix. The figure shows the number of meta-features retained after each step. For visualization purposes, a meta-feature advances to the next step whenever it passes a test for at least one of the studied performance gaps.

*Table 1.* Retained robust associations. $\rho$ denotes univariate Spearman correlation with dataset level performance gap and Sign cons. is the bootstrap sign consistency. $q_{\mathrm{BH}}$ values are Benjamini Hochberg FDR adjusted $p$ values. The adjusted coefficient and sign consistency are post-screening covariate checks with fixed dataset controls.

| Comparison | Feature | $n$ | $\rho$ (95% CI) | $q_{BH}$ | Sign cons. | Adj. coef. (95% CI) | Adj. sign cons. |
|---|---|---|---|---|---|---|---|
| Non-TFM / TFM | attr_ent.skewness | 50 | $-0.510$ $[-0.709, -0.231]$ | .043 | 1.00 | $-0.536$ $[-0.899, -0.202]$ | .996 |
| TabICLv2 / TabPFN-2.6 | attr_conc.median | 50 | $-0.510$ $[-0.725, -0.258]$ | .043 | 1.00 | $-0.546$ $[-0.864, -0.248]$ | .998 |

1992). This two criterion rule guards against nominally significant but directionally unstable associations. We report percentile bootstrap confidence intervals (CIs) for all retained features. Finally, we perform a multivariate covariate adjustment check by fitting a rank-based linear model including the retained feature and fixed dataset controls and verify that the adjusted coefficient retains the univariate direction with bootstrap sign consistency $> 0.90$.

**Control Features.** Previous work often started association discovery from scratch although certain aspects of a dataset are well-known and well understood. Therefore, analogous to social science studies, we define a set of meta-features as control features: log_n, log_d, d_over_n, cat_fraction, and feature_missing_fraction. Those features were defined because they are known to influence models, e.g. sample and feature counts influence foundation models (Hollmann et al., 2023; Qu et al., 2026), and CatBoost (Prokhorenkova et al., 2018) is known to excel over other models at categorical feature handling (Tschalzev et al., 2024).

**Predictive evaluation.** We ask whether meta-features can predict the gap for datasets not seen during training, using leave-one-dataset-out cross-validation. In each fold, one dataset is held out and the remaining datasets form the training set. We evaluate five feature sets: (1) fixed controls only, (2) all meta-features, (3) controls plus all meta-features, and, if available, (4) robust meta-features and (5) controls plus robust meta-features. As meta-predictor model, we compare TabPFN-2.6 (Hollmann et al., 2023), which requires no hyperparameter tuning and handles missing values natively, against our mean/majority baselines. We report sign accuracy for the predicted gap direction and additionally mean absolute error (MAE) between predicted and observed held-out gaps in the appendix. For MAE, the mean baseline predicts the average gap observed in the training datasets. For sign accuracy, the majority baseline predicts the model

family that wins more often in the training datasets. These baselines serve as references for assessing whether meta-features provide additional out-of-sample routing information.

## 4. Results

We focus on three comparisons that isolate different routing questions: (1) non-TFM vs. TFM models, (2) TabICLv2 vs. TabPFN-2.6, and (3) NNs vs. tree-based models. We also compare newer versions of the TabICL and TabPFN families to older ones in Appendix B. Figure 2 summarizes the predictive result. The numerical sign-accuracy values with confidence intervals and MAE results are reported in Appendix B.2.

**Most meta-feature associations disappear under robust screening.** Figure 1 summarizes the filtering process of the association analysis. Although the general meta-features show several high correlations across the main comparisons, the combination of accounting for multi-hypothesis testing and bootstrap sign-consistency filtering leaves only two retained associations (Table 1). This shows that under the TabArena protocol, most apparent global meta-feature explanations are not stable enough to report as robust model-family effects.

**The NN vs. tree and Non-TFM vs. TFM provide little routing signal.** For NN vs. tree, no general meta-feature survives the robust association screen, and the predictive evaluation shows no compensating sign-routing signal from the larger feature sets (Figure 2). Thus, the meta-features used here do not recover a reliable NN/tree routing rule, extending the finding of (Tschalzev et al., 2025). For Non-TFM vs. TFM, skewness of attribute entropy is retained as a robust association (Table 1): attribute entropy measures the Shannon entropy of each predictive attribute, and its skewness describes whether the entropy distribution across

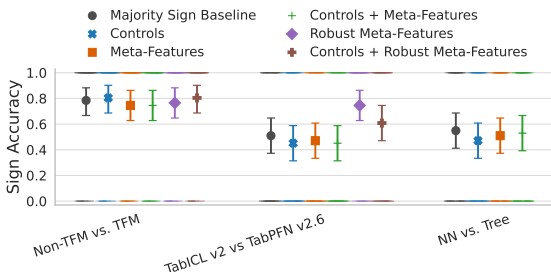

*Figure 2.* Leave-one-dataset-out sign accuracy for the three main comparisons. Points show how often each feature set predicts the correct held-out gap direction. Error bars show 95% bootstrap confidence intervals. Higher values indicate better routing accuracy.

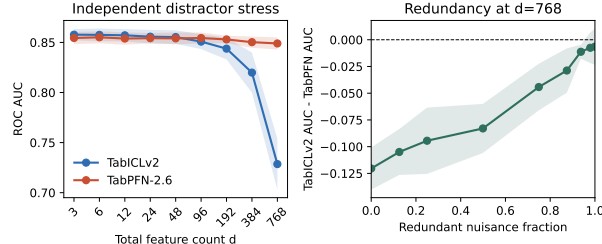

*Figure 3.* Nuisance-feature controls. Left: model AUC as independent nuisance dimensions are added to a fixed three-feature signal task. Right: AUC gap under fixed $d = 768$. The independent nuisance columns are progressively converted into near-duplicates. Shading denotes $\pm 1$ standard deviation over seeds.

attributes is balanced or dominated by unusually low- or high-entropy attributes. However, using this feature as a router does not improve sign accuracy over the majority-sign baseline (Figure 2). We therefore treat the Non-TFM/TFM association as descriptive rather than decision-useful.

**TabICLv2 vs. TabPFN-2.6 shows the strongest meta-feature signal.** For TabICLv2 vs. TabPFN-2.6, the retained meta-feature is `attr_conc.median`, the median pairwise feature concentration across all columns. This meta-feature measures typical mutual predictability between features, so higher values indicate stronger redundancy among dataset columns. Since the association is negative under our gap convention, we hypothesize that datasets with more redundant features favor TabICLv2, whereas datasets with less redundant and more independently informative features favor TabPFN-2.6. Appendix B.3 and Figure 3 probe this interpretation with controlled nuisance-feature ablations. Unlike the broad Non-TFM/TFM feature, this single retained feature improves both held-out gap magnitude and held-out gap direction, with the sign-accuracy gain being visually the strongest effect in Figure 2 .

# 5. Discussion and Conclusion

The main lesson from our analysis is that global meta-features provide limited evidence for model-family routing on TabArena. We deliberately separate association from prediction: the screening stage asks which meta-features remain stable after multiple-hypothesis-testing control, while the leave-one-dataset-out evaluation tests whether they generalize to unseen datasets.

**Trade-off between evaluation quality and benchmarking time.** The most important limitation of our study is sample size. TabArena's 51 manually curated datasets, combined with hundreds of meta-features, create a $d >> n$ problem. Many predictors are correlated summaries of related dataset properties, so redundancy filtering, false discovery control, and bootstrap sign checks reduce over-

interpretation but cannot create independent evidence. The lack of robust meta-features therefore admits two interpretations: (1) Reliably detecting global meta-feature patterns requires more datasets, given the heterogeneity of tabular data. (2) TabArena prioritized extensive but reliable evaluation on few datasets, whereas prior benchmarks used many. This trade-off between evaluation quality and dataset count means current benchmarks may not support useful meta-feature conclusions.

**Other limitations.** Several additional assumptions constrain our claims. The results depend on the TabArena model implementations, validation-based family selection, normalized error definition, and family definitions used here, and foundation-model behavior may change as models, training corpora, and context-size limits evolve. All meta-features used here are hand-engineered and mostly model-agnostic. Some computationally expensive ones could not be computed for every dataset within our budget, so the analysis uses the available dataset-feature pairs rather than a complete matrix. In the predictive evaluation, the robust-feature predictor sets use features retained by the full association analysis rather than features selected within each leave-one-dataset-out fold, so therefore these results assess whether the reported robust associations carry predictive signal rather than providing a fully nested estimate of out-of-sample routing performance.

**Conclusion** Across the main TabArena comparisons, global meta-features provide at most weak routing signal under our protocol. The NN vs. tree and Non-TFM vs. TFM comparisons yield no practically useful predictive association. The TabICLv2 vs. TabPFN-2.6 comparison produces one robust, predictively useful signal in median feature concentration which results in a concrete, falsifiable hypothesis about when these two models diverge. Future work will reveal which interpretation holds: the TabArena datasets may be too heterogeneous for the global meta-features studied here to capture reliable model-family differences, or 51 datasets may simply be too few to surface stable patterns.

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

# A. Methodology Details

## A.1. TabArena Benchmark

**Overview.** TabArena (Erickson et al., 2025) is a curated, living benchmark for supervised machine learning on tabular data. It provides 51 independently verifiable IID tasks spanning binary classification, multiclass classification, and regression. Datasets were selected through a manual curation process intended to reduce common repository artifacts, including data leakage, mislabeled targets, duplicate instances, and ill-defined prediction targets. Each task is stored as an OpenML dataset (Vanschoren et al., 2014) and linked to a standardized evaluation protocol using 10-repeat 3-fold cross-validation, yielding up to 30 predefined train/validation/test splits per dataset. All splits are fixed prior to model evaluation so that every method is evaluated on identical data partitions.

**Evaluation protocol.** For each split and each evaluated method, TabArena records both validation and test predictions. Method configurations are evaluated under three subtypes: `default` (fixed hyperparameters), `tuned` (validation-based HPO), and `tuned_ensemble` (post-HPO ensembling). Error metrics depend on the task type: log-loss for classification and root mean squared error for regression. The normalized score used throughout this paper (Equation (2)) maps each method's raw test error to a $[0, 1]$ scale anchored to the per-split best and median error across all evaluated methods.

**Methods.** The benchmark evaluates tree-based methods (CatBoost (Prokhorenkova et al., 2018), LightGBM (Ke et al., 2017), XGBoost (Chen & Guestrin, 2016), Extra-Trees (Geurts et al., 2006), RandomForest (Breiman, 2001), EBM (Lou et al., 2013), PerpetualBoost (ML, 2025)), NNs (FastAI (Erickson et al., 2020), MNCA (Ye et al., 2024), NN-Torch (Erickson et al., 2020), RealMLP (Holzmüller et al., 2024), TabM (Gorishniy et al., 2025)), classical baselines (KNN, logistic regression), and foundation models that perform in-context inference without task-specific gradient updates: TabPFN v2 (Hollmann et al., 2023), TabPFN-2.6 (Grinsztajn et al., 2026), RealTabPFN v2.5 (Garg et al., 2025), TabICL (Qu et al., 2025), TabICLv2 (Qu et al., 2026), TabDPT (Ma et al., 2024), and TabSTAR (Arazi et al., 2025). We compare individual method-family portfolios only, as defined in Appendix 3.

**Dataset summary.** Table 2 lists all 51 datasets used in this paper. Datasets span ten application domains, with business and finance tasks being most frequent. Dataset sizes range from 748 to 150,000 instances and from 5 to 1,777 features. All 51 datasets are publicly available via their respective OpenML task identifiers. See Erickson et al. (2025) for the complete task ID mapping and per-dataset curation notes.

*Table 2.* All 51 TabArena datasets used in this paper, grouped by task type. Type: Bin. = binary classification, Multi. = multiclass classification, Reg. = regression. $n$: number of instances. $d$: total number of features. %cat: fraction of categorical features (rounded to one decimal place).

| Dataset | Type | $n$ | $d$ | %cat |
|---|---|---|---|---|
| Blood Transfusion | Bin. | 748 | 5 | 20.0 |
| Diabetes | Bin. | 768 | 9 | 11.1 |
| Anneal | Multi. | 898 | 39 | 84.6 |
| QSAR Fish Toxicity | Reg. | 907 | 7 | 0.0 |
| Credit-G | Bin. | 1,000 | 21 | 66.7 |
| Maternal Health Risk | Multi. | 1,014 | 7 | 14.3 |
| Concrete Strength | Reg. | 1,030 | 9 | 0.0 |
| QSAR Biodeg | Bin. | 1,054 | 42 | 14.3 |
| Health Insurance | Reg. | 1,338 | 7 | 42.9 |
| Website Phishing | Multi. | 1,353 | 10 | 100.0 |
| Fitness Club | Bin. | 1,500 | 7 | 57.1 |
| Airfoil Self-Noise | Reg. | 1,503 | 6 | 16.7 |
| Used Fiat 500 | Reg. | 1,538 | 8 | 12.5 |
| MIC | Multi. | 1,699 | 112 | 84.8 |
| Good Customer | Bin. | 1,723 | 14 | 64.3 |
| Marketing Campaign | Bin. | 2,240 | 26 | 34.6 |
| Hazelnut Contaminant | Bin. | 2,400 | 31 | 3.2 |
| Seismic Bumps | Bin. | 2,584 | 16 | 25.0 |
| Splice | Multi. | 3,190 | 61 | 100.0 |
| Bioresponse | Bin. | 3,751 | 1,777 | 0.1 |
| HIVA Agnostic | Multi. | 3,845 | 1,618 | 100.0 |
| Student Dropout | Multi. | 4,424 | 37 | 48.6 |
| Churn | Bin. | 5,000 | 20 | 25.0 |
| QSAR-TID-11 | Reg. | 5,742 | 1,025 | 0.0 |
| Polish Bankruptcy | Bin. | 5,910 | 65 | 1.5 |
| Wine Quality | Reg. | 6,497 | 13 | 7.7 |
| Taiwanese Bankruptcy | Bin. | 6,819 | 95 | 1.1 |
| NATICUSdroid | Bin. | 7,491 | 87 | 100.0 |
| COIL 2000 | Bin. | 9,822 | 86 | 4.7 |
| Bank Churn | Bin. | 10,000 | 11 | 45.5 |
| HELOC | Bin. | 10,459 | 24 | 4.2 |
| JM1 | Bin. | 10,885 | 22 | 4.5 |
| E-Commerce Shipping | Bin. | 10,999 | 11 | 45.5 |
| Online Shoppers | Bin. | 12,330 | 18 | 44.4 |
| Coupon Recommendation | Bin. | 12,684 | 25 | 88.0 |
| Miami Housing | Reg. | 13,776 | 16 | 6.3 |
| HR Job Change | Bin. | 19,158 | 13 | 76.9 |
| Houses | Reg. | 20,640 | 9 | 0.0 |
| Superconductivity | Reg. | 21,263 | 82 | 0.0 |
| Credit Card Default | Bin. | 30,000 | 24 | 16.7 |
| Amazon Empl. Access | Bin. | 32,769 | 10 | 100.0 |
| Bank Marketing | Bin. | 45,211 | 14 | 57.1 |
| Food Delivery Time | Reg. | 45,451 | 10 | 30.0 |
| Physiochem. Protein | Reg. | 45,730 | 10 | 0.0 |
| KDD Cup 09 | Bin. | 50,000 | 213 | 18.3 |
| Diamonds | Reg. | 53,940 | 10 | 30.0 |
| Diabetes 130US | Bin. | 71,518 | 48 | 83.3 |
| APS Failure | Bin. | 76,000 | 171 | 0.6 |
| SDSS17 | Multi. | 78,053 | 12 | 25.0 |
| Airline Satisfaction | Bin. | 129,880 | 22 | 77.3 |
| Give Me Credit | Bin. | 150,000 | 11 | 9.1 |

## A.2. Error Normalization

For a given split $s$ of dataset $D$, let $e_{\min}(D, s)$ denote the minimum (best) test error over all evaluated methods, and let $e_{\mathrm{med}}(D, s)$ denote their median test error. The normalized error of a method with raw error $e$ is

$$\tilde{e}(D, s) = \mathrm{clip}\left(\frac{e - e_{\min}(D, s)}{r(D, s)}, \; 0, \; 1\right), \qquad (2)$$

where $r(D, s) = \max\big(e_{\mathrm{med}}(D, s) - e_{\min}(D, s), \; \varepsilon\big)$ and $\varepsilon = 10^{-5}$.

A value of 0 corresponds to matching the best method on that split. A value of 1 corresponds to matching the per-split median. The denominator clipping prevents division by near-zero spreads and ensures the score lies in $[0, 1]$. Because both $e_{\min}$ and $e_{\mathrm{med}}$ are computed from the same pool of methods on each $(D, s)$ task, they serve as data-driven, task-adaptive anchors rather than fixed external baselines.

The same normalization is applied separately to validation errors for within-family model selection. Concretely, equation (2) is applied to the raw validation error $e^{\mathrm{val}}$ per $(D, s)$, again anchored to the global per-split method pool, yielding $\tilde{e}^{\mathrm{val}}(D, s)$. The best representative of each family on split $s$ is the method that minimizes $\tilde{e}^{\mathrm{val}}(D, s)$ within the family. Its held-out test performance is then evaluated via the separately computed $\tilde{e}(D, s)$ from equation (2).

## A.3. Family Definitions

The three routing comparisons (rows 1–3) use all 51 datasets. The three version-to-version checks (rows 4–6) are restricted to the dataset subsets applicable to each model, as noted in Appendix B.

# B. Additional Results

## B.1. Association Screening

Tables 4 to 9 list the top nominally significant meta-feature associations for each of the six comparisons. We define these as associations with raw Spearman $p < 0.05$ and report at most ten features per comparison. The reported $q_{\mathrm{BH}}$ values are Benjamini Hochberg FDR adjusted $p$ values from the corresponding feature screen. The criterion $q_{\mathrm{BH}} < 0.05$ is the multiplicity control requirement used in the reporting rule.

Note, that not all comparisons use all 51 datasets. TabPFN v2 (Hollmann et al., 2025) is restricted in TabArena to datasets with at most 10,000 training samples, at most 500 features, and at most 10 classes for classification tasks. These restrictions make 33 of the 51 datasets theoretically applicable to TabPFN v2. TabICL v1 (Qu et al., 2025) is restricted in TabArena to classification tasks with at most

*Table 3.* Model families used in the six comparisons. Each family pools all default and tuned variants of the listed methods. Within each comparison block, the first row corresponds to family A and the second row to family B.

| Family | Methods |
|---|---|
| MLP-based (A) | FastAI, MNCA (GPU), NN-Torch, RealMLP (GPU), TabM (GPU) |
| Tree-based (B) | CatBoost, LightGBM, XGBoost, EBM, PerpetualBoost, ExtraTrees, RandomForest |
| non-TFM (A) | FastAI, MNCA (GPU), NN-Torch, RealMLP (GPU), TabM (GPU), Cat-Boost, LightGBM, XGBoost, EBM, PerpetualBoost, ExtraTrees, KNN, LR, RandomForest |
| TFM (B) | TabDPT (GPU), TabSTAR, TabI-CLv2, TabICL (GPU), TabPFN v2 (GPU), TabPFN-2.6, RealTabPFN v2.5 |
| TabICLv2 (A) TabPFN-2.6 (B) | TabICLv2 TabPFN-2.6 |
| RealTabPFN v2.5 (A) TabPFN-2.6 (B) | RealTabPFN v2.5 TabPFN-2.6 |
| TabPFN v2 (A) TabPFN-2.6 (B) | TabPFN v2 (GPU) TabPFN-2.6 |
| TabICL v1 (A) TabICLv2 (B) | TabICL (GPU) TabICLv2 |

100,000 training samples and at most 500 features. These restrictions make 36 of the 51 datasets theoretically applicable to TabICL v1. This excludes all 13 regression datasets and two classification datasets that exceed the feature dimensionality limit. The reduced sample sizes in these two comparisons lower statistical power relative to the four comparisons with full benchmark coverage.

## B.2. Predictive Evaluation

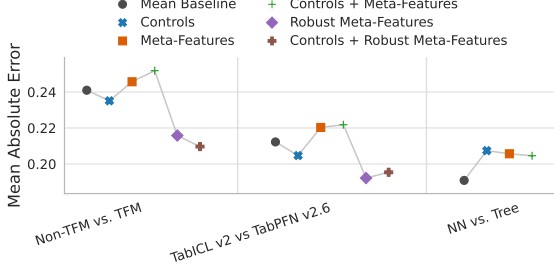

*Figure 4.* Leave-one-dataset-out MAE. Points show the held-out gap prediction error. Lower values indicate better prediction.

**More meta-features lead to overfitting.** Across the all-feature and controls-plus-all-feature variants generally fail to improve over the mean-gap baseline for MAE and do not deliver reliable sign routing (Figures 4 and 2). The evi-

dence therefore contrasts previous high-dimensional global meta-feature routing as a promising research direction, and indicates that hypothesis-driven analysis based on selected meta-features is more promising.

Additionally, Tables 10 to 15 report the TabPFN leave-one-dataset-out results for all six comparisons under the predictive protocol defined in Section 3. The feature-set labels follow the same definitions: MF abbreviates meta-features, and Robust MF denotes the retained robust association features from the screening procedure. In the baseline row, MAE uses the training-set mean predictor and sign accuracy uses the training-set majority-sign predictor. Confidence intervals are percentile bootstrap intervals over held-out datasets using 5000 bootstrap resamples.

*Table 4.* Top nominally significant meta-feature associations for NN vs. Tree. Rows show up to 10 features with raw Spearman $p < 0.05$. $q_{BH}$ is reported for context.

| Feature | $n$ | $\rho$ | $p$ | $q_{BH}$ |
|---|---|---|---|---|
| pymfe_sparsity.histogram.5 | 45 | -0.387 | .009 | .867 |
| pymfe_skewness.min | 42 | -0.376 | .014 | .867 |
| pymfe_attr_ent.range | 50 | 0.337 | .017 | .867 |
| pymfe_mad.histogram.4 | 46 | 0.350 | .017 | .867 |
| high_cardinality_fraction | 50 | 0.328 | .020 | .867 |
| pymfe_attr_ent.sd | 50 | 0.327 | .020 | .867 |
| pymfe_sparsity.range | 45 | 0.342 | .022 | .867 |
| pymfe_sparsity.sd | 45 | 0.331 | .026 | .872 |
| high_corr_pair_fraction | 44 | -0.332 | .028 | .872 |
| pymfe_sd.min | 47 | -0.307 | .036 | .969 |

*Table 5.* Top nominally significant meta-feature associations for Non-TFM vs. TFM. Rows show up to 10 features with raw Spearman $p < 0.05$. $q_{BH}$ is reported for context.

| Feature | $n$ | $\rho$ | $p$ | $q_{BH}$ |
|---|---|---|---|---|
| pymfe_attr_ent.skewness | 50 | -0.510 | < .001 | .043 |
| pymfe_attr_ent.histogram.9 | 50 | 0.405 | .004 | .309 |
| pymfe_cor.histogram.6 | 43 | 0.427 | .004 | .309 |
| pymfe_sparsity.skewness | 44 | 0.421 | .004 | .309 |
| pymfe_sparsity.histogram.0 | 45 | 0.395 | .007 | .347 |
| pymfe_attr_conc.median | 50 | 0.375 | .007 | .347 |
| pymfe_sparsity.median | 45 | -0.365 | .014 | .360 |
| pymfe_kurtosis.histogram.4 | 43 | 0.370 | .015 | .360 |
| pymfe_sparsity.quantiles.1 | 45 | -0.362 | .015 | .360 |
| pymfe_cor.histogram.8 | 43 | 0.369 | .015 | .360 |

*Table 6.* Top nominally significant meta-feature associations for TabICLv2 vs. TabPFN-2.6. Rows show up to 10 features with raw Spearman $p < 0.05$. $q_{BH}$ is reported for context.

| Feature | $n$ | $\rho$ | $p$ | $q_{BH}$ |
|---|---|---|---|---|
| pymfe_attr_conc.median | 50 | -0.510 | < .001 | .043 |
| pymfe_attr_conc.iq_range | 50 | -0.419 | .002 | .325 |
| pymfe_nr_cor_attr | 47 | -0.415 | .004 | .325 |
| pymfe_cor.median | 43 | -0.414 | .006 | .325 |
| pymfe_median.histogram.9 | 46 | -0.390 | .007 | .325 |
| pymfe_t_mean.iq_range | 46 | -0.389 | .008 | .325 |
| pymfe_iq_range.median | 46 | -0.373 | .011 | .325 |
| pymfe_attr_ent.kurtosis | 50 | 0.356 | .011 | .325 |
| pymfe_median.histogram.0 | 46 | 0.369 | .012 | .325 |
| pymfe_mad.histogram.9 | 46 | -0.369 | .012 | .325 |

*Table 7.* Top nominally significant meta-feature associations for RealTabPFN v2.5 vs. TabPFN-2.6. Rows show up to 10 features with raw Spearman $p < 0.05$. $q_{BH}$ is reported for context.

| Feature | $n$ | $\rho$ | $p$ | $q_{BH}$ |
|---|---|---|---|---|
| pymfe_attr_conc.histogram.5 | 50 | 0.379 | .007 | .613 |
| pymfe_attr_conc.histogram.4 | 50 | 0.378 | .007 | .613 |
| pymfe_skewness.histogram.2 | 42 | 0.407 | .008 | .613 |
| pymfe_attr_ent.range | 50 | 0.357 | .011 | .613 |
| pymfe_sd.histogram.2 | 47 | -0.365 | .012 | .613 |
| feature_missing_fraction | 51 | 0.345 | .013 | .613 |
| pymfe_attr_conc.histogram.6 | 50 | 0.338 | .016 | .633 |
| pymfe_attr_ent.sd | 50 | 0.331 | .019 | .633 |
| pymfe_var.histogram.8 | 47 | -0.338 | .020 | .633 |
| d_over_n | 51 | -0.304 | .030 | .845 |

*Table 8.* Top nominally significant meta-feature associations for TabPFN v2 vs. TabPFN-2.6. Rows show up to 10 features with raw Spearman $p < 0.05$. $q_{BH}$ is reported for context.

| Feature | $n$ | $\rho$ | $p$ | $q_{BH}$ |
|---|---|---|---|---|
| pymfe_attr_ent.histogram.0 | 33 | -0.446 | .009 | .977 |
| pymfe_skewness.skewness | 32 | 0.424 | .015 | .977 |
| pymfe_var.histogram.6 | 33 | 0.404 | .020 | .977 |
| pymfe_min.histogram.8 | 33 | 0.399 | .021 | .977 |
| pymfe_attr_conc.histogram.4 | 33 | 0.377 | .030 | .977 |
| pymfe_min.histogram.7 | 33 | 0.376 | .031 | .977 |
| pymfe_attr_ent.histogram.8 | 33 | -0.366 | .036 | .977 |
| pymfe_attr_conc.histogram.7 | 33 | 0.356 | .042 | .977 |

*Table 9.* Top nominally significant meta-feature associations for TabICL v1 vs. TabICLv2. Rows show up to 10 features with raw Spearman $p < 0.05$. $q_{BH}$ is reported for context.

| Feature | $n$ | $\rho$ | $p$ | $q_{BH}$ |
|---|---|---|---|---|
| pymfe_attr_ent.histogram.5 | 36 | -0.454 | .005 | .708 |
| effective_rank | 32 | 0.444 | .011 | .708 |
| pymfe_cor.histogram.1 | 32 | -0.441 | .011 | .708 |
| pymfe_cor.histogram.2 | 32 | -0.439 | .012 | .708 |
| participation_ratio | 32 | 0.427 | .015 | .708 |
| pymfe_eigenvalues.histogram.8 | 33 | 0.413 | .017 | .708 |
| pymfe_cor.histogram.3 | 32 | -0.405 | .021 | .708 |
| pymfe_cov.histogram.5 | 33 | -0.391 | .025 | .708 |
| log_d | 36 | 0.371 | .026 | .708 |
| pymfe_cor.histogram.7 | 32 | -0.384 | .030 | .708 |

*Table 10.* TabPFN leave-one-dataset-out predictive evaluation for Non-TFM vs. TFM. Values in brackets are 95% bootstrap confidence intervals across held-out datasets.

| Predictor | $n$ | $n_{\text{pred}}$ | MAE (95% CI) | Sign accuracy (95% CI) |
|---|---|---|---|---|
| Baseline | 51 | 0 | 0.241 [0.192, 0.296] | 0.784 [0.667, 0.882] |
| Controls | 51 | 5 | 0.235 [0.187, 0.287] | 0.804 [0.686, 0.902] |
| MF | 51 | 277 | 0.242 [0.194, 0.297] | 0.765 [0.647, 0.882] |
| Controls + MF | 51 | 282 | 0.248 [0.199, 0.302] | 0.745 [0.627, 0.863] |
| Robust MF | 51 | 1 | 0.217 [0.170, 0.270] | 0.765 [0.647, 0.882] |
| Controls + Robust MF | 51 | 6 | 0.212 [0.162, 0.266] | 0.784 [0.667, 0.902] |

*Table 11.* TabPFN leave-one-dataset-out predictive evaluation for TabICLv2 vs. TabPFN-2.6. Values in brackets are 95% bootstrap confidence intervals across held-out datasets.

| Predictor | $n$ | $n_{\text{pred}}$ | MAE (95% CI) | Sign accuracy (95% CI) |
|---|---|---|---|---|
| Baseline | 51 | 0 | 0.212 [0.160, 0.270] | 0.510 [0.373, 0.647] |
| Controls | 51 | 5 | 0.205 [0.155, 0.262] | 0.451 [0.314, 0.588] |
| MF | 51 | 277 | 0.218 [0.163, 0.281] | 0.451 [0.314, 0.588] |
| Controls + MF | 51 | 282 | 0.217 [0.163, 0.278] | 0.451 [0.314, 0.588] |
| Robust MF | 51 | 1 | 0.195 [0.148, 0.246] | 0.765 [0.647, 0.882] |
| Controls + Robust MF | 51 | 6 | 0.193 [0.144, 0.247] | 0.647 [0.510, 0.784] |

*Table 12.* TabPFN leave-one-dataset-out predictive evaluation for NNs vs. tree-based models. Values in brackets are 95% bootstrap confidence intervals across held-out datasets.

| Predictor | $n$ | $n_{\text{pred}}$ | MAE (95% CI) | Sign accuracy (95% CI) |
|---|---|---|---|---|
| Baseline | 51 | 0 | 0.191 [0.152, 0.231] | 0.549 [0.412, 0.686] |
| Controls | 51 | 5 | 0.207 [0.167, 0.249] | 0.471 [0.333, 0.608] |
| MF | 51 | 277 | 0.210 [0.168, 0.252] | 0.471 [0.333, 0.608] |
| Controls + MF | 51 | 282 | 0.208 [0.166, 0.251] | 0.490 [0.353, 0.627] |

*Table 13.* TabPFN leave-one-dataset-out predictive evaluation for RealTabPFN v2.5 vs. TabPFN-2.6. Values in brackets are 95% bootstrap confidence intervals across held-out datasets.

| Predictor | $n$ | $n_{\text{pred}}$ | MAE (95% CI) | Sign accuracy (95% CI) |
|---|---|---|---|---|
| Baseline | 51 | 0 | 0.151 [0.114, 0.194] | 0.647 [0.510, 0.784] |
| Controls | 51 | 5 | 0.148 [0.114, 0.187] | 0.608 [0.471, 0.745] |
| MF | 51 | 277 | 0.162 [0.123, 0.205] | 0.569 [0.431, 0.706] |
| Controls + MF | 51 | 282 | 0.164 [0.125, 0.207] | 0.588 [0.451, 0.725] |

*Table 14.* TabPFN leave-one-dataset-out predictive evaluation for TabPFN v2 vs. TabPFN-2.6. Values in brackets are 95% bootstrap confidence intervals across held-out datasets.

| Predictor | $n$ | $n_{\text{pred}}$ | MAE (95% CI) | Sign accuracy (95% CI) |
|---|---|---|---|---|
| Baseline | 33 | 0 | 0.203 [0.138, 0.273] | 0.818 [0.667, 0.939] |
| Controls | 33 | 4 | 0.229 [0.162, 0.299] | 0.818 [0.667, 0.939] |
| MF | 33 | 282 | 0.219 [0.156, 0.290] | 0.818 [0.667, 0.939] |
| Controls + MF | 33 | 286 | 0.221 [0.156, 0.293] | 0.818 [0.667, 0.939] |

*Table 15.* TabPFN leave-one-dataset-out predictive evaluation for TabICL v1 vs. TabICLv2. Values in brackets are 95% bootstrap confidence intervals across held-out datasets.

| Predictor | $n$ | $n_{\text{pred}}$ | MAE (95% CI) | Sign accuracy (95% CI) |
|---|---|---|---|---|
| Baseline | 36 | 0 | 0.202 [0.160, 0.246] | 0.889 [0.778, 0.972] |
| Controls | 36 | 5 | 0.202 [0.155, 0.251] | 0.889 [0.778, 0.972] |
| MF | 36 | 277 | 0.219 [0.173, 0.268] | 0.889 [0.778, 0.972] |
| Controls + MF | 36 | 282 | 0.219 [0.172, 0.269] | 0.889 [0.778, 0.972] |

### B.3. Interpreting median attribute concentration

The retained feature `attr_conc.median` is the median Goodman and Kruskal concentration coefficient over ordered pairs of input attributes, computed after PyMFE's categorical transformation of the data. It is target free and measures typical association between features: high values indicate that one input column is often predictive of another input column, whereas low values indicate less mutually predictable columns. On real tables, high values therefore tend to coincide with redundant columns and hence lower effective dimensionality. In these synthetic controls, continuous columns are discretized by PyMFE before computing `attr_conc`, so independent Gaussian columns have a small finite sample baseline rather than exactly zero. For wide synthetic tables, the reported value uses PyMFE's `max_attr_num=12` sampling and averages over ten sampling seeds.

**Nuisance directions.** All controls use seeds $s \in \{0, \ldots, 4\}$, 400 training examples, and 1200 test examples. The base task samples three signal features $x_1, x_2, x_3 \sim \mathcal{N}(0, 1)$ and generates labels according to

$$y \sim \text{Bernoulli}\left(\sigma(1.5x_1 - 1.1x_2 + 0.8x_3 + 0.6x_1x_2)\right).$$

The independent nuisance branch appends $\mathcal{N}(0, 1)$ columns to obtain $d \in \{3, 6, 12, 24, 48, 96, 192, 384, 768\}$. The redundant nuisance branch fixes $d = 768$ and replaces independent nuisance columns with copies of one nuisance latent coordinate. We use the same copy operation in both redundancy controls: a latent coordinate $v$ contributes one exact observed column, and any additional copies are $v + 0.03\epsilon$, with $\epsilon \sim \mathcal{N}(0, 1)$. This changes redundancy while preserving the supervised label function for the nuisance control.

Figure 3 and Table 16 summarize the pattern. Independent nuisance directions keep `attr_conc.median` near its low baseline and open a large TabICLv2 deficit, while redundant nuisance copies raise `attr_conc.median` and reduce that deficit.

**Informative directions.** The nuisance experiment leaves open a simpler explanation: TabICLv2 may only struggle because the added high dimensional columns are irrelevant. We therefore repeat the stress test with target relevant feature directions. In the independent branch, every observed feature is informative, so $d = k$ and no nuisance columns are present. The $k = 3$ condition is the same base task as above. For larger $k$, additional independent observed coordinates enter the label function through a normalized random linear component. Formally, for $k > 3$ we draw $x = (x_1, x_2, x_3, z_1, \ldots, z_{k-3})$, with all coordinates sampled independently from a standard normal distribution, and define

$$\eta_3(x) = 1.5x_1 - 1.1x_2 + 0.8x_3 + 0.6x_1x_2,$$
$$r_k(x) = \eta_3(x) + \alpha\, s_3\, \frac{w_k^\top z}{\|w_k\|_2}.$$

The final logit is centered and rescaled as

$$\eta_k(x) = s_3 \frac{r_k(x) - \bar{r}_k}{\text{sd}(r_k)}, \qquad y \sim \text{Bernoulli}(\sigma(\eta_k(x))),$$

where $s_3 = \text{sd}(\eta_3(x))$, $\bar{r}_k$, and $\text{sd}(r_k)$ are computed on the combined generated training and test sample. For each

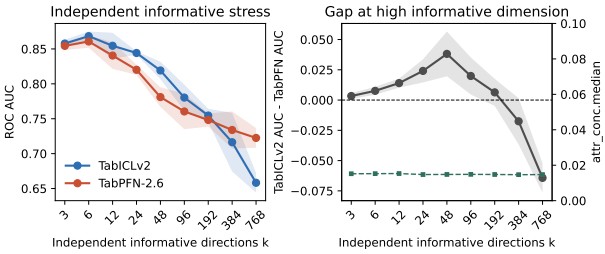

*Figure 5.* Independent informative direction stress test. Every observed feature is target relevant, so $d = k$. TabICLv2 remains competitive for low and moderate $k$, but its AUC decreases faster than TabPFN 2.6 at large independent informative dimensionality.

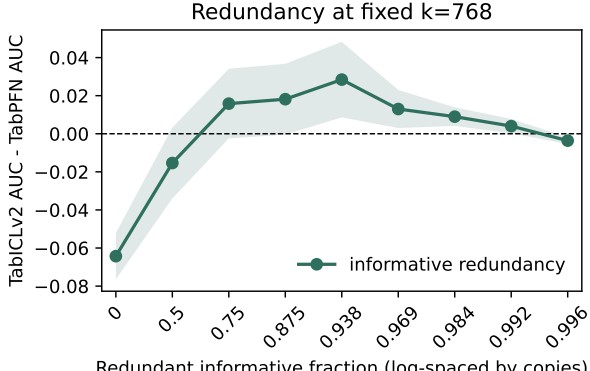

*Figure 6.* Fixed-$k = 768$ informative redundancy control. The horizontal axis shows the redundant informative fraction, with positions spaced logarithmically by the number of copies per latent factor. Reducing the number of distinct target relevant latent factors moves the AUC gap back toward TabICLv2.

*Table 16.* Representative summary for the four controls. Values are means over five paired seeds. The AUC gap is $\text{AUC}_{\text{TabICLv2}}$ minus $\text{AUC}_{\text{TabPFN 2.6}}$.

| Control | Setting | `attr_conc` | TabICLv2 AUC | TabPFN 2.6 AUC | AUC gap |
|---|---|---|---|---|---|
| Nuisance | independent $d = 3$ | 0.015 | 0.858 | 0.854 | 0.003 |
| Nuisance | independent $d = 768$ | 0.015 | 0.729 | 0.849 | -0.120 |
| Nuisance | redundant $d = 768$ | 0.862 | 0.831 | 0.837 | -0.006 |
| Informative | independent $k = 3$ | 0.015 | 0.858 | 0.854 | 0.003 |
| Informative | independent $k = 768$ | 0.015 | 0.658 | 0.723 | -0.064 |
| Informative | redundant $k = 768, m = 48$ | 0.015 | 0.809 | 0.780 | 0.028 |

copy control is a mechanism check rather than a direct validation of `attr_conc.median`: because redundancy is spread across several independent informative groups, the sampled median concentration remains close to the low baseline.

Taken together, the controls support a narrow interpretation. `attr_conc.median` is not a measure of target relevance, but an unsupervised signal for whether raw columns behave more like many independent directions or redundant copies. The benchmark correlation is consistent with TabICLv2 being more sensitive to large effective feature dimensionality, especially for nuisance directions, while redundancy reduces the relative deficit.

seed and $k$, the entries of $w_k$ are drawn as random signs times $\text{Uniform}(0.5, 1.5)$ magnitudes and then normalized by $\|w_k\|_2$. We use $\alpha = 1$ throughout. Thus, increasing $k$ spreads comparable scale target signal over more independent directions rather than simply increasing the raw logit magnitude. At $k = 3$, we use the unmodified base task above.

Figure 5 shows that target relevance reduces but does not remove the high dimensional stress. As $k$ grows, both models lose AUC, and the gap remains unfavorable to TabICLv2 (Table 16).

Finally, we fix the observed informative dimensionality at $k = 768$ and vary the number $m \in \{3, 6, 12, 24, 48, 96, 192, 384, 768\}$ of distinct target relevant latent factors. Labels are generated from the $m$ latent factors using the same logit construction above, and the copy operation maps them to 768 observed informative columns, with $768/m$ columns per latent factor. Thus $m = 768$ matches the independent $k = 768$ endpoint, while smaller $m$ creates redundant informative copies. This informative

