# OpenReview forum: "Explaining Tabular Foundation Model Differences Through Meta-Features"
_ICML.cc/2026/Workshop/FMSD — FMSD @ ICML 2026 Poster_

### Official Review · Reviewer_HfjJ · 2026-05-15
**A Careful but Somewhat Limited Study of Meta-Feature-Based Routing for Tabular Models**

**Rating:** 5
**Confidence:** 5

**Review:**

This paper revisits whether dataset meta-features can explain or predict performance differences between model families on tabular data. Using TabArena results, the authors study several comparisons, including non-foundation models vs. tabular foundation models, TabICLv2 vs. TabPFN-2.6, and neural networks vs. tree-based models. They compute a large set of dataset-level meta-features and apply a fairly strict analysis pipeline, including redundancy pruning, FDR correction, bootstrap sign-consistency checks, and leave-one-dataset-out prediction. The main conclusion is that global meta-features provide only weak and limited routing signal. Most apparent associations disappear after robust statistical filtering, and meta-feature-based predictors generally do not improve much over simple baselines.

Strengths:
The paper asks a relevant question for the workshop. As tabular foundation models become stronger, practitioners need to know when a TFM should be preferred over classical or non-foundation tabular models. Studying whether dataset meta-features can support this kind of model routing is therefore well aligned with the theme of foundation models for structured data.

I appreciate the careful distinction between association and prediction. Many papers report correlations between dataset properties and model performance, but this paper explicitly checks whether such features generalize to held-out datasets. The use of simple mean and majority-sign baselines is also important, since it prevents overclaiming the usefulness of meta-feature predictors.


Areas for Improvement:
The main limitation is that the paper’s conclusions are constrained by the small number of datasets. TabArena has only 51 datasets, while the number of candidate meta-features is large. The authors acknowledge this issue, but it still makes the negative conclusion hard to interpret: it is unclear whether global meta-features are truly not useful, or whether the study is simply underpowered.

The paper is also somewhat limited in terms of practical takeaways. The conclusion that meta-features are mostly not robust enough is useful, but the paper does not provide much guidance on what kind of alternative routing signal might work better. For a workshop paper, it would be helpful to include a more forward-looking discussion of possible replacements, such as learned dataset embeddings, landmarking features, small-budget probing, or model-performance-based warm starts.

Another limitation is that the retained meta-features are not really interpreted. For example, the paper finds that median attribute concentration is useful for TabICLv2 vs. TabPFN-2.6, but does not analyze why this feature matters or what kind of datasets it corresponds to. This makes the result harder to use or build on.

Detailed Comments:
1. The paper should be more cautious in wording the main conclusion. Given the small sample size, I would frame the result as “current global meta-features are not sufficient on TabArena” rather than a broader statement about meta-features for tabular data in general.

2. It would be useful to explain the retained robust features in more intuitive terms. What does “attribute entropy skewness” or “median attribute concentration” mean for a real dataset? What kinds of datasets have high or low values? This would make the findings more interpretable.

3. The leave-one-dataset-out evaluation is a good idea, but the robust-feature predictor sets are selected using the full association analysis. The authors mention this limitation, but it should be made clearer in the main methodology or results section, since it affects how one should interpret the predictive numbers.

4. The paper could include additional baselines for routing. For example, landmarking features based on running a few cheap models or using a small validation subset may be more predictive than static meta-features. Comparing against such baselines would make the routing analysis more complete.

5. The family definitions are important for the conclusions. In particular, the Non-TFM vs. TFM comparison pools many quite different methods on both sides. The authors should discuss whether the weak signal may come from heterogeneity within each family, rather than from a general failure of meta-features.

6. The paper uses TabPFN as the meta-predictor for leave-one-dataset-out gap prediction. This is reasonable, but it would be helpful to report whether simpler regressors, such as ridge regression, random forests, or linear models on the selected features, lead to similar conclusions.

7. Since the work is about model routing, sign accuracy is arguably more important than MAE. The paper could emphasize this more clearly and discuss cases where MAE improves but routing accuracy does not.

---

### Official Review · Reviewer_9NHP · 2026-05-20
**This paper could lead to interesting discussions at the workshop**

**Rating:** 6
**Confidence:** 3

**Review:**

## Summary
The paper "Revisiting Metafeatures to Explain Model Differences on Tabular Data" uses dataset metafeatures to predict the difference between tabular foundation models and other model classes. After an elaborate cleaning procedure, only 2 features exhibit a correlation between the order of the meta-feature value and the order of the performance differences. In predictive settings, this correlation did not result in improved model selection.

## Strengths
* Interesting research questions: we do not understand when tabular foundation models would outperform other models. Explaining this using meta-features would be interesting.
* Elaborate statistical analysis of feature correlation. The authors ensure to correct for multiple testing correction

## Areas for Improvement
* The paper works under the assumption that there is a monotonous relation between a meta-feature and the difference in performance. I believe this is too restrictive, and the authors should consider other, more powerful analysis as well.
* The problem under study is called "algorithm selection" and sometimes also referred to as "model selection". There is a rich literature on this topic (see Joaquin Vanschoren's survey on meta-learning in the AutoML book), and reporting the mean predictive performance of the meta-model (selector) is misleading. Ideally, one reports the selection regret, as making a wrong decision on a dataset where the two models under choice behave the same isn't an issue. However, this will not explain model differences. But if the models are very similar on a dataset, what difference should we explain?
* Line 73 (left) mentions omitted baselines in related work. However, from the text it is unclear, and the paper should clearly state what baselines are missing in related work.
* Line 156 (right) mention a "predictive plot", but I do not know what this is. There should be a reference to this plot in the text.

All in all, this is a borderline paper. But since my criticism regarding presentation can be fixed for the workshop, I think the paper will lead to interesting discussions. Therefore, I think accepting it would be fine. However, I strongly encourage the authors to look into more complex statistical models or to use machine learning models to perform algorithm selection to allow taking into account meta-feature interactions.

---

### Official Review · Reviewer_nHBo · 2026-05-21
**Review of the paper: "Revisiting Metafeatures to Explain Model Differences on Tabular Data"**

**Rating:** 4
**Confidence:** 3

**Review:**

**Strengths:**

- This paper clearly proposed two interesting questions: It asks (1) whether meta-features explains performance gaps and (2) whether it predicts the gap on unseen data.
- The screening pipeline is statistically rigorous. It combines FDR control, a bootstrap sign-consistency rule, and a covariate check.
- The paper’s experiments are controlled and the benchmarking process is sound. TabArena is a good choice in this context.
- I like that this paper is transparent about its findings. It reports failure modes/associations (which is missing in a lot of TFM papers), and openly lists its own limitations.

**Weakness:**

- As mentioned in Section 5’s limitation, the robust features were chosen using all datasets, so the held-out dataset is not truly unseen. This could result in non-nested robust-feature predictor and potentially data leakage.
- As mentioned in “Trade-off between evaluation quality and benchmarking time,” 51 datasets are too few (d >> n) and might result in low statistical significance in the results.
- The prediction test relies on a single meta-predictor, TabPFN. Using only TabPFN makes it impossible to know if the weak signal is real or model-specific.
- The two result findings sit on a thin margin. Both barely pass the FDR cutoff, and their predictive gain over the baseline is not clearly separated.

**Questions and Suggestions:**

- Notation: Equation (1) uses normalized errors A and B, but these are defined only later in Appendix A.2 via Equation (2), so a key term is used before it is defined (Eq. 1; Appendix A.2). Re-order the equations or bring them both to the main text would help.
- Address this limitation would help strengthen the conclusion: “In the predictive evaluation, the robust-feature predictor sets use features retained by the full association analysis rather than features selected separately within each leave-one-dataset-out training fold, so those results assess whether the reported robust associations carry predictive signal rather than providing a fully nested estimate of out-of-sample routing performance.”
- Use more TFMs than TabPFN as the meta-predictor.
- Typos (?) Is the “ProtoBoost” in Table 3 the same as “PerpetualBoost” in Section A.1’s “Methods”
- Reference need to be cleaned up:
    - For McElfresh et al., 2023 : Missing venue in the reference
    - Some of the arxiv citations have URL and some does not. Please be consistent in referencing work.